# Contribution of Ribbon-Structured SiO_2_ Films to AlN-Based and AlN/Diamond-Based Lamb Wave Resonators

**DOI:** 10.3390/s23146284

**Published:** 2023-07-10

**Authors:** Mohammed Moutaouekkil, Jérémy Streque, Othmane Marbouh, El Houssaine El Boudouti, Omar Elmazria, Philippe Pernod, Olivier Bou Matar, Abdelkrim Talbi

**Affiliations:** 1University of Lille, CNRS, Centrale Lille, Univ. Polytechnique Hauts-de-France, UMR 8520-IEMN, F-59000 Lille, France; othmane.marbouh@centralelille.fr (O.M.); philippe.pernod@centralelille.fr (P.P.); olivier.boumatar@iemn.univ-lille1.fr (O.B.M.); abdelkrim.talbi@univ-lille.fr (A.T.); 2LPMR, Faculté des Sciences, Université Mohammed I, Oujda 60000, Morocco; elboudouti@yahoo.fr; 3Université de Lorraine, CNRS, IJL UMR 7198, F-54000 Nancy, France; jeremy.streque@gmail.com (J.S.); omar.elmazria@univ-lorraine.fr (O.E.)

**Keywords:** S_0_ Lamb-wave mode, K^2^, AlN/SiO_2_ bi-layer, SiO_2_ ribbons, TCF

## Abstract

New designs based on S_0_ Lamb modes in AlN thin layer resonating structures coupled with the implementation of structural elements in SiO_2_, are theoretically analyzed by the Finite Element Method (FEM). This study compares the typical characteristics of different interdigital transducer (IDTs) configurations, involving either a continuous SiO_2_ cap layer, or structured SiO_2_ elements, showing their performance in the usual terms of electromechanical coupling coefficient (K^2^), phase velocity, and temperature coefficient of frequency (TCF), by varying structural parameters and boundary conditions. This paper shows how to reach temperature-compensated, high-performance resonator structures based on ribbon-structured SiO_2_ capping. The addition of a thin diamond layer can also improve the velocity and electromechanical coupling coefficient, while keeping zero TCF and increasing the solidity of the membranes. Beyond the increase in performance allowed by such resonator configurations, their inherent structure shows additional benefits in terms of passivation, which makes them particularly relevant for sensing applications in stern environments.

## 1. Introduction

In recent years, surface acoustic wave (SAW) devices have been extensively used for very numerous applications such as mobile and wireless communications [1,2,3]. These devices are most often based on piezoelectric substrates whose phase velocity is usually low, like lithium niobate and quartz. It makes the manufacturing process more difficult, and restrains their increase in frequency of use for future telecommunication standards. High phase-velocity materials were then introduced in order to increase their operating frequency, including diamond, silicon, and sapphire [4,5,6,7,8,9].

Recently, various electro-acoustic devices based on aluminum nitride thin film technology have also been proposed for frequency control and sensing applications at high temperature [10,11,12,13,14,15,16,17]. Some of these devices are based on bulk acoustic waves (BAW), following either thin-film bulk acoustic resonator technologies (FBAR), or solidly mounted resonator technologies (SMR) [18,19,20]. Their operating frequency depends on the thickness of the piezoelectric films involved in their fabrication, contrary to other technologies, whose operating frequencies mostly depend on the lateral dimensions of their patterned electrodes, which paves the way towards even higher frequencies. However, FBAR and SMR technologies must rely on extremely accurate layer engineering in order to fulfill their requirements in terms of frequency and performance. Beyond BAW devices, other devices relying on AlN-based Lamb wave resonators (LWR) were also recently proposed, as they combine the advantages of both SAW and BAW technologies [21,22].

The AlN-based piezoelectric materials present several advantages, including high phase velocity [6,7], good thermal conductivity [11], low dielectric properties [23], and high elastic stiffness coefficients [6,23]. Moreover, the AlN-based piezoelectric materials are largely used in the MEMS process and are compatible with the complementary metal–oxide semiconductor (CMOS) [24]. Furthermore, other materials have been widely studied such as Gallium Nitride, Zinc Oxide and Zirconate Titanate [25,26,27]. According to the literature, it is apparent that a compromise exists between material choice, manufacturing process, device performance, and CMOS compatibility.

In the literature, various structures have been investigated such as AlN/Si, AIN/Diamond, AlN/Sapphire and AlN/SiC [28,29,30,31], in order to increase the electromechanical coupling coefficient (K^2^). Aluminum nitride was then considered to be a good candidate for the elaboration of Lamb wave resonators based on S_0_ mode, by virtue of its high phase velocity near 10 km/s, large K^2^, and high quality factor (Q), as well as its small temperature coefficient of frequency and strong resistance to harsh environments.

Moreover, this structure can also be operated in high temperature conditions. Indeed, significant research efforts have been focused on AlN-based piezoelectric devices for high temperature applications [32,33,34,35,36,37]. However, frequency stability over temperature remains a major challenge for AlN-based resonators [18,38]. Several studies were carried out for their temperature compensation through the addition of a silicon oxide layer to the resonating structures, benefitting from the positive TCF of SiO_2_ [39,40]. Structures based on SiO_2_/AlN membranes were then proposed, achieving thermal compensation at the cost of a lower K^2^ coefficient. In this study, we consider an original structure which allows a better trade-off between TCF cancellation and resonator performance for S_0_ Lamb-wave resonating structures. A recent work [41] was previously proposed by the authors of this current study, where preliminary results of this original structure were presented for SiO_2_ ribbons for the design of zero-TCF resonators based on S_0_ Lamb-wave mode. We then perform a complete study on the introduction of SiO_2_ ribbons deposited on AlN and AlN/Diamond bilayer membranes, as an alternative to the continuous SiO_2_ thin films usually introduced in such structures.

## 2. Design and Methods

A series of multilayered structures based on AlN membranes combined with additional SiO_2_ films have been compared in the first part of this study. Different configurations were considered depending on the position, thickness and shape of the SiO_2_ layers, and on the topology of the interdigital transducers (IDTs), as shown in Figure 1. Three electrode topologies were proposed: A is equipped with a single set of electrodes on the AlN membrane’s top side (IDT/AlN), while B introduces a continuous metal layer on its back side (IDT/AlN/Metal). The third topology (C) proposes two sets of electrodes on each side of the AlN membrane (IDT/AlN/IDT).

For each of these electrode topologies, two different membrane configurations have been discussed. The first configuration is based on the addition of a continuous SiO_2_ layer, on top of the AlN electrode structures. The second membrane configuration proposes a novel SiO_2_ structure made of ribbons deposited over the IDTs. For the calculation of dispersion curves, unit cells corresponding to one wavelength of the structure have been defined, with the choice of symmetric continuity conditions at their boundaries. The dispersion curves are calculated numerically based on the general Partial Derivative Equation (PDE) interface, using Comsol Multiphysics ^®^ Software based on FEM. The approach consists of using the following dynamic relationship through an anisotropic electro-elastic membrane, while neglecting electric current density, as well as change of electric charges, and body forces, namely:(1)$$\left\{\begin{array}{c} \frac{\partial^{2}u_{i}}{\partial t^{2}}=\frac{1}{\rho_{0}}\frac{\partial T_{\mathrm{ij}}}{\partial x_{j}}\;\\ \;\frac{\partial D_{i}}{\partial x_{i}}=0,\; \end{array}\right.\;$$
with the following constitutive equations: (2)$$\left\{\begin{array}{c} T_{\mathrm{ij}}=C_{\mathrm{ijkl}}\frac{\partial u_{l}}{\partial x_{k}}+e_{\mathrm{kij}}\frac{\partial \emptyset_{e}}{\partial x_{k}},\;\\ D_{i}=e_{\mathrm{kji}}\frac{\partial u_{j}}{\partial x_{k}}+\varepsilon_{\mathrm{ij}}\frac{\partial \emptyset_{e}}{\partial x_{j}},\; \end{array}\right.$$where e_ikl_, C_ijkl_ and ε_jk_ denote the effective piezoelectric constants, the second order elastic constants, and the dielectric permittivity tensor, respectively.

Inserting Equations (1) and (2) into Equation (3), the governing equations of plane wave propagation in 2D electro-elastic membrane can be written as:(3)$$\left\{\begin{array}{c} \rho_{0}\frac{\partial^{2}u_{i}}{\partial t^{2}}{=C}_{ijkl}\frac{\partial^{2}u_{k}}{\partial x_{l}\partial x_{j}}+e_{kij}\frac{\partial^{2}\emptyset_{e}}{\partial x_{k}\partial x_{j}},\;\\ {0=e}_{jkl}\frac{\partial^{2}u_{k}}{\partial x_{l}\partial x_{j}}-\varepsilon_{jk}\frac{\partial^{2}\emptyset_{e}}{\partial x_{k}\partial x_{j}}.\; \end{array}\right.$$

In order to calculate the electromechanical coupling coefficient and phase velocity of the bi-layer membrane, we seek a solution in the form of plane waves:(4)$$u_{i}=u_{i}^{0}e^{-{\mathrm{jk}}_{x}x}e^{-{\mathrm{jk}}_{y}y}e^{-j\omega t}.$$

The system of equations is then transformed into an eigenvalue problem, which is given by:(5)$$\lambda^{2}e_{a}{\bar{u}}^{0}-{\lambda d}_{a}{\bar{u}}^{0}+\nabla \Gamma =F,$$
where the eigenvalue is linked to the pulsation by the relation:(6)λ=−jω,
with
$$d_{a}=0,e_{a}=\left[\begin{array}{cccc} {-\rho }_{0} & 0 & 0 & 0\\ 0 & {-\rho }_{0} & 0 & 0\\ 0 & 0 & {-\rho }_{0} & 0\\ 0 & 0 & 0 & 0 \end{array}\right],\;\Gamma =\left[\begin{array}{ccc} T_{11}^{0} & T_{12}^{0} & T_{13}^{0}\\ T_{21}^{0} & T_{22}^{0} & T_{23}^{0}\\ T_{31}^{0} & T_{32}^{0} & T_{33}^{0}\\ D_{1}^{0} & D_{2}^{0} & D_{3}^{0} \end{array}\right]\;and\;F=\left[\begin{array}{c} jk_{x}T_{11}^{0}+{jk_{y}T}_{13}^{0}\\ jk_{x}T_{12}^{0}+{jk_{y}T}_{22}^{0}\\ jk_{x}T_{13}^{0}+{jk_{y}T}_{23}^{0}\\ jk_{x}D_{1}^{0}+{jk_{y}D}_{2}^{0} \end{array}\right]$$

For each configuration, the width of the unit cell is set to a wavelength value of λ = 4 µm; this typically fits operation in the GHz frequency range. 

In this study, the influence of the electrodes was neglected, therefore they do not appear in the simulated structures. The TCF coefficient of resonators is related to the decrease in the elastic constants, and to the thermal expansion of the materials involved in the structure of the membranes. The temperature coefficients of the material constants of AlN and SiO_2_ used in the calculation are taken from the literature [42]. We have summarized the physical properties used in this work (as well as diamond, as introduced later in this paper) in Table 1.

Parametric studies have then been carried out. The simulations have then been run for a variable AlN membrane thickness (h_AlN_), for a kh_AlN_ (relative AlN thickness) spanning from 0 to 6. The relative thickness of the SiO_2_ films has also been investigated as another parameter for this parametric study, for each of its possible implementations as continuous layers or ribbons.

The aim of the simulations was to optimize the structures in order to maximize the K^2^ coefficient, while cancelling the TCF. The study was focused on the S_0_ Lamb-wave mode, which presents higher K^2^ coefficients than the A_0_ mode. As a first step, the optimal SiO_2_ thickness leading to the cancellation of TCF has been determined for either continuous layers or ribbons of silicon dioxide. Then, dispersion curves including phase velocity, K^2^, and TCF, have been retrieved and compared for these optimal SiO_2_ thicknesses. 

The K^2^ coefficient for the Lamb-wave mode was determined from the difference between the phase velocities for two boundary conditions on the potential: open-circuit and short-circuit [39]. The K^2^ coefficient is defined as: (7)$$K^{2}=2\frac{V_{0}-V_{m}}{V_{0}}$$
where V_0_ and V_m_ are the phase velocities for open-circuit and short-circuit, respectively.

## 3. Results and Discussion

### 3.1. TCF and Phase Velocity in SiO_2_/AlN Structures

The temperature coefficient of frequency was first determined for AlN membranes equipped with either continuous SiO_2_ film or SiO_2_ ribbons, for various silicon dioxide thicknesses. As explained before, the width of the unit cells (corresponding to λ) has been set to 4 µm. Figure 2a shows the simulation results for both temperature compensation strategies. When continuous SiO_2_ films are introduced in the structure, the TCF dispersion curve shows a fast transition from positive TCF values to negative ones, leaving little room for temperature compensation. Then, Figure 2b clearly shows one, two or even three compensation points depending on the thickness of the SiO_2_ layer. We can also notice the existence of a plateau for relative AlN thicknesses (1.7 < kh_AlN_ < 2.4). For this design, the optimal SiO_2_ thickness would be 540 nm (relative thickness (kh_SiO2_ = 0.85)).

As shown in Figure 3, the proposed implementation of structured SiO_2_ capping layers can strongly influence the dispersion curves and performances of such resonator structures, making it possible to adjust the boundary loads (including mass and electrical loads). This large dispersion of phase velocity is particularly visible for low relative AlN thicknesses (kh_AlN_ below 2.6). In this velocity domain, we noticed that the dispersion curve for the AlN membrane with SiO_2_ ribbons (red solid line) is positioned between those of SiO_2_/AlN bi-layer membrane (blue solid line), and AlN membrane (black solid line).

On the contrary, the TCF dispersion curve obtained for AlN membranes equipped with SiO_2_ ribbons (red solid line) makes it possible to have several compensation points, and gives the possibility of adjusting the SiO_2_ thickness in order to obtain a lower dependence on kh_AlN_. This would offer more leeway for the design of sensors based on this structure, especially if an additional sensing layer has to be implemented in the structure. The corresponding AlN thicknesses would also be lower than for continuous AlN films, giving access to higher operating frequencies. In this case, the optimal SiO_2_ thickness for temperature compensation is 540 nm (kh_SiO2_ = 0.85), which is also thinner than in the case of SiO_2_/AlN bi-layers.

### 3.2. Electromechanical Coupling Coefficients in SiO_2_/AlN Structures

In this part of the work, the SiO_2_ layer thickness has been optimized in order to modify the shape of the dispersion curve, while avoiding the appearance of the stopband frequency. Dispersion curves and the K^2^ coefficient depend strongly on the sensitivity of the acoustic wave to the surface charge which can be identified by the difference in velocity of charged and uncharged surfaces. In the region where there is a maximum dispersion of velocity for which we can obtain optimal sensitivity, this region is equivalent to a maximum difference between phase velocity (v_p_) and group velocity (v_g_), i.e., v_p_–v_g_, with v_g_ = ∂ω/∂k, k is the wavevector and ω the angular frequency. This hypothesis was mentioned by the authors in reference [44] associated with the propagation characteristics of Lamb waves in the AlN/Diamond bi-layer membrane. This property can be used to explain any change in the behavior of the K^2^ coefficient in both cases: the SiO_2_/AlN bi-layer structure and the AlN membrane with SiO_2_ ribbons. In the same way, we can explain the variation in TCF using the same hypothesis for both cases.

As shown before, another silicon dioxide thickness presents a strong interest in this study, this time for AlN membranes equipped with SiO_2_ ribbons: 540 nm (relative thickness of 0.85, for λ = 4 µm). Thus, similar dispersion curves have been generated for this SiO_2_ thickness, as shown in Figure 4, for both continuous silicon dioxide films and ribbons. As expected, the SiO_2_/AlN bi-layer membrane does not exhibit much leeway for TCF compensation, with a strong TCF variation in the neighborhood of the unique compensation point at about kh_AlN_ = 2.64 as shown in Figure 4a (red dotted up arrow). From these results, the electrode topology (IDT/AlN/IDT) reaches K^2^ of around 3% for the SiO_2_/AlN bi-layer membrane, whereas, for the other electrode topologies such as (IDT/AlN) and (IDT/AlN/Metal), they exhibit, respectively, K^2^ close to 1.3% and 2.2% for kh_AlN_ = 2.64 (Figure 4a).

As expected, the AlN membranes equipped with SiO_2_ ribbons benefit most from this SiO_2_ thickness. They exhibit good aptitudes for TCF compensation, and for relative AlN thicknesses (1.7 < kh_AlN_ < 2.4), while showing higher K^2^ coefficients, up to 3% and 4% for two electrode topologies: (IDT/AlN/Metal) and (IDT/AlN/IDT), respectively (see Figure 4b). Such a resonator structure leads to even higher K^2^ coefficients than with the previously proposed structures with 640 nm thick SiO_2_ ribbons [41]. It also keeps an edge in terms of design robustness, since both K^2^ and TCF share the same optimal AlN thickness (kh_AlN_ = 2.1), and remain rather steady around this value. The cell width is fixed to 4µm and the length and thickness of the SiO_2_ ribbons are fixed to 2 µm and 640 nm, respectively. There is no experience at this stage because unfortunately nowadays our team does not have the experimental means to carry out the manufacturing of AlN membranes.

We can explain the change in the electromechanical coupling coefficient for the three structural configurations by the fact that the interdigital transducers generate lateral and vertical electric fields in the AlN membrane. In addition, the membrane with SiO_2_ ribbons and IDT/AlN metal topology makes it possible to modify the distribution of the electric field: the vertical component becomes preponderant due to the low dielectric permittivity of SiO_2_ compared to that of the AlN membrane. The lateral electric component is very strong compared with the vertical electric component for high thicknesses. However, this topology presents a vertical electric component which becomes much larger than the lateral electric component for low thicknesses.

### 3.3. Phase Velocity in SiO_2_/AlN/Diamond Structures

The previously studied SiO_2_/AlN structures already exhibit strong performances in terms of TCF cancellation and electromechanical coupling. In this section, a diamond layer has been introduced underneath the previously studied resonator structures, in order to improve their high frequency performance. The various multilayer structures and their electrode topologies are summed up in Figure 5. As before, the IDTs’ electrical conductivity was neglected.

Similarly to the previous study of SiO_2_/AlN structures, the optimal thicknesses of SiO_2_ and diamond were determined in order to get a near-zero TCF plateau when using a ribbon-structured SiO_2_ layer. The chosen SiO_2_ thickness was 400 nm, while the diamond thickness was set to 500 nm. These values correspond to relative thicknesses of 0.6 and 0.75, respectively.

Figure 6 shows an increase in phase velocity when adding the diamond layer for both continuous and ribbon-like SiO_2_ film configurations. The AlN/Diamond bi-layer membrane with SiO_2_ ribbons (red solid line) shows a phase velocity close to 12000 m/s, which is larger than the velocities reached for SiO_2_/AlN/Diamond (blue solid line). We can also notice the dispersion curve of the structure equipped with ribbons (blue solid line) which approximates to the dispersion curve of the AlN/Diamond bi-layer membrane (black solid line). 

### 3.4. Electromechanical Coupling Coefficient of SiO_2_/AlN/Diamond Structures

In this section, the K^2^ and TCF coefficients of S_0_ Lamb-wave mode for the two layer configurations introduced previously were calculated: the first one corresponds to a multilayered structure of SiO_2_/AlN/Diamond, while the second one consists of an AlN/Diamond bi-layer structure equipped with SiO_2_ ribbons. The material constants of the diamond and the temperature coefficients used in the calculation are taken from the literature (See Table 1) [43]. Figure 7 presents the dispersion curves for K^2^ and TCF obtained from the simulations. One can see that SiO_2_/AlN/Diamond multilayered structures only offer one compensation point, for a relative thickness of 2.07 (red dotted up arrow). Moreover, the electrode topologies based on (IDT/AlN) only show residual electromechanical coupling with K^2^ coefficients close to 1%. The best electrode topology, based on (IDT/AlN/IDT), only exhibits a mild coupling coefficient of 3.7%.

Finally, as illustrated in Figure 7b, the addition of diamond underneath the AlN/SiO_2_ ribbons structure still enables TCF cancellation over a large range of relative AlN thickness (1.5 < kh_AlN_ < 2.32), similarly to the previously studied AlN/SiO_2_ ribbon structures. The presence of diamond offers even larger K^2^ coefficients, reaching 5.2% for the electrode topology (IDT/AlN/IDT), whereas for other electrode topologies such as (IDT/AlN) and (IDT/AlN/Metal), they exhibit, respectively, K^2^ close to 2% and 3.8% (Figure 7b). 

In summary, these new structures benefit from the low permittivity of their additional diamond layer, compared to AlN. This changes the electric boundary conditions, and thus amplifies the gradient of the electric field in the AlN membrane. Consequently, the phase velocity could be increased, as well as the electromechanical coupling coefficient, leading to even more performant zero-TCF resonators. The TCF compensation would also be reached through a low kh_AlN_ of 2.07 (red dotted up arrow) for SiO_2_/AlN/Diamond multilayered structures, and even authorize a wide range of relative thickness when the SiO_2_ layer is structured into ribbons. 

The proposed structures (Ribbon-Structured SiO_2_ Films to AlN-based and AlN/Diamond-based) can operate in the frequency domain around 2.3 GHz, with an electromechanical coupling coefficient close to 4% and 5.2%, phase velocity of 9200 and 9500 m/s, and a temperature coefficient of frequency close to 0 ppm/ °C.

We have summarized in Table 2 the advantages and the disadvantages for different device structures in the range of kh_AlN_ [0.6–2.6].

The calculation of the quality factor requires several geometric parameters of resonator, namely number of pairs of the IDT fingers, and the number of reflectors installed. In this work, we have taken into consideration only the mechanical energy loss from the resonator, knowing that the actual quality factor (Q) depends on different energy loss mechanisms.

As is well known, the effective coupling coefficient $k_{\mathrm{eff}}^{2}$ is defined by the approximation formula [45]:$$k_{\mathrm{eff}}^{2}=\frac{\pi^{2}}{4}\frac{f_{p}-f_{s}}{f_{p}}$$
where *f*_p_ and *f*_s_ are the parallel frequency and series frequency, respectively.

Figure 8 below shows the evolution of the admittance as a function of the relative frequency using an AlN membrane with SiO_2_ ribbons for three electrode topologies: IDT/AlN/Metal, IDT/AlN and IDT/AlN/IDT. One can see that the electrode topology IDT/AlN/IDT configuration presents a large effective coupling coefficient $k_{\mathrm{eff}}^{2}\;$≈ 4.9% compared to other electrode topologies.

Figure 9 shows the $k_{\mathrm{eff}}^{2}$ for three electrode topologies IDT/AlN/Metal, IDT/AlN and IDT/AlN/IDT using an AlN membrane with SiO_2_ ribbons of thickness h_SiO2_ = 540 nm. We noticed a similar behavior to that of K^2^ presented previously in the paper but with a slight increase in $k_{\mathrm{eff}}^{2}$.

For the fabrication process of the AlN-based resonator, a 4 μm thick AlN layer is deposited by reactive sputtering on high-resistivity silicon wafers. The subsequent definition of the IDT is achieved through photolithography, with a bi-layer photoresist process (LOR 3A + AZ 1518). Then, an Al thin film (150 nm thick) is deposited by electron beam evaporation or thermal evaporation, followed by a lift-off process based on usual strippers, to free the top IDTs.

## 4. Conclusions

In this study, we have proposed new structured platforms for high-velocity, strong-coupling and zero-TCF resonators. In the first part, structures based on the association of AlN membranes and SiO_2_ ribbons covering the IDTs were proposed for use in S_0_-mode Lamb wave resonators. Such an AlN/SiO_2_ ribbon structure exhibits high phase velocity, and a large K^2^ (exceeding 4%). This structure can lead to good temperature compensation for an AlN thickness lower than in the case of a SiO_2_/AlN bi-layer membrane. In the second part, the addition of a diamond layer to the membrane structure was proposed, in order to reach even higher velocities and coupling coefficients. The lower permittivity of diamond helps to confine the electric field into the AlN layer, while its mechanical properties authorize larger velocities. The maximal free velocity in the membranes can thus be increased up to 12,000 m/s, while a coupling coefficient of 5.2% is predicted for the optimal structure, through a wide range of temperature compensation. The studied structures are also well suited for reliable operation at very high temperature, and thus pave the way towards radiofrequency devices and sensors withstanding harsh environments. Additionally, the outstanding mechanical properties of diamond makes it possible to improve the quality factor and the rigidity of the resonators.

## Figures and Tables

**Figure 1 sensors-23-06284-f001:**
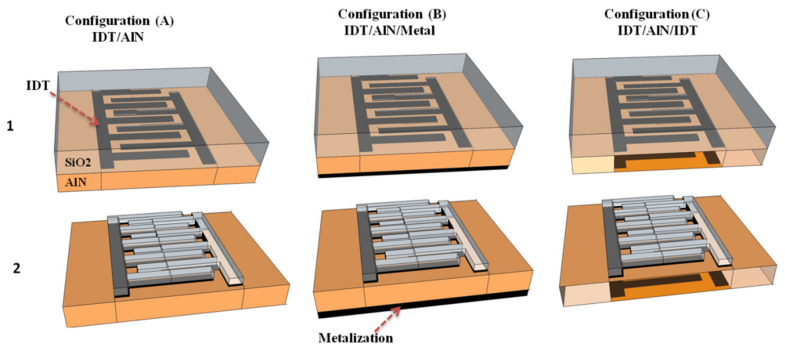
Illustration of the unit cells of the studied multilayered structures for three electrode topologies (**A**–**C**): bilayer membrane with continuous SiO_2_ film on top (1) of the AlN, and AlN membrane equipped with SiO_2_ ribbons over the IDTs (2).

**Figure 2 sensors-23-06284-f002:**
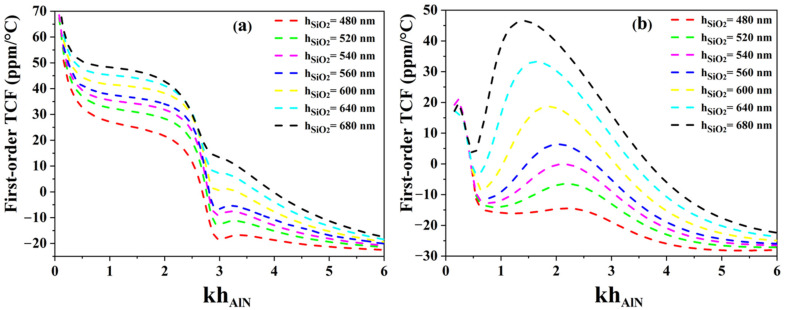
Comparison of TCF dispersion curves for (**a**) SiO_2_/AlN bi-layer membrane and (**b**) AlN membrane equipped with SiO_2_ ribbons.

**Figure 3 sensors-23-06284-f003:**
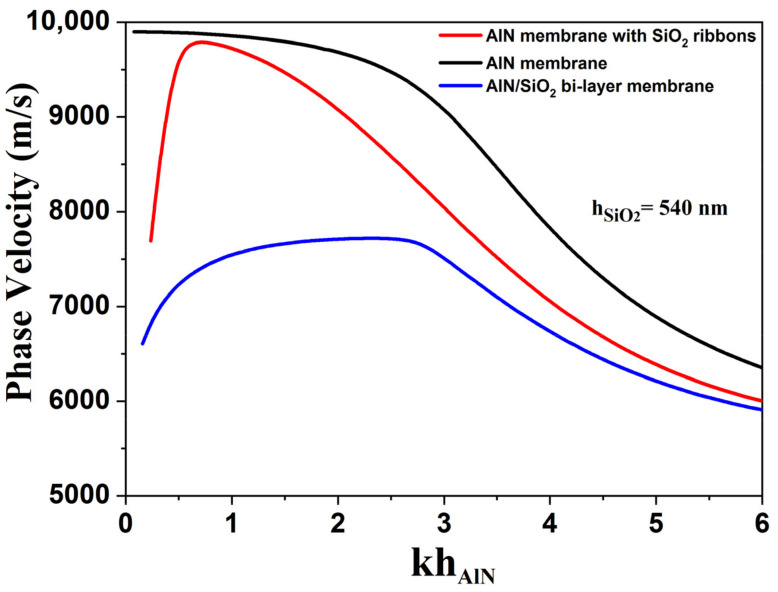
Comparison of the phase velocity dispersion curves for AlN membrane (black solid line), SiO_2_/AlN bi-layer membrane (blue solid line), and AlN membrane with SiO_2_ ribbons (red solid line).

**Figure 4 sensors-23-06284-f004:**
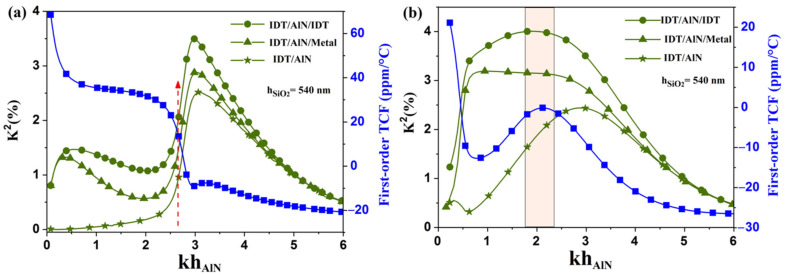
Comparison of K^2^ and TCF dispersion curves for (**a**) SiO_2_/AlN bi-layer membrane, and (**b**) AlN membrane with SiO_2_ ribbons for h_SiO2_ = 540 nm. Green curves correspond to K^2^ coefficient and blue curve correspond to TCF.

**Figure 5 sensors-23-06284-f005:**
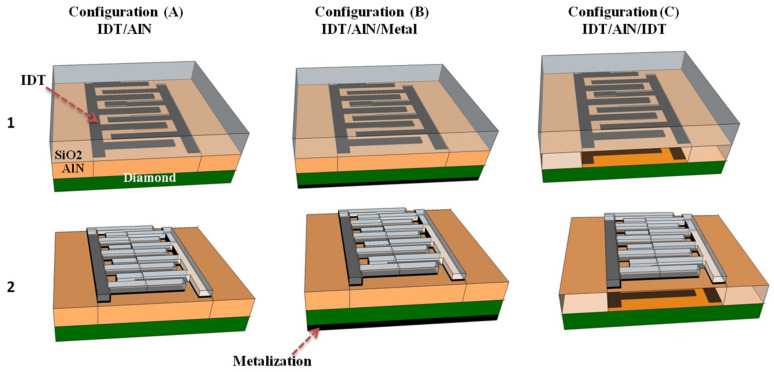
Illustration of the unit cells of the studied multilayered structures for three electrode topologies (**A**–**C**): multilayer membrane with continuous SiO_2_ film on top (1) of the AlN/Diamond membrane, and AlN/Diamond bi-layer membrane equipped with SiO_2_ ribbons over the IDTs (2).

**Figure 6 sensors-23-06284-f006:**
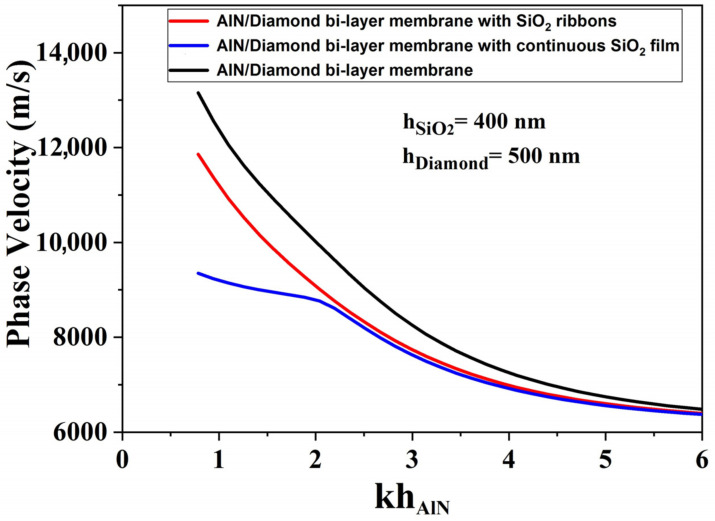
Comparison of the phase velocity dispersion curves for AlN/Diamond bi-layer membrane with SiO_2_ ribbons (red solid line), SiO_2_/AlN/Diamond (blue solid line), and AlN/Diamond bi-layer membrane (black solid line).

**Figure 7 sensors-23-06284-f007:**
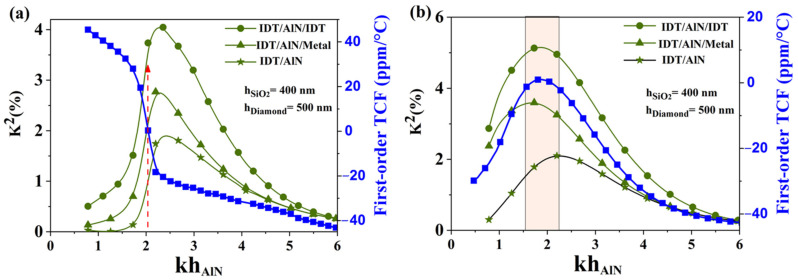
Comparison of K^2^ and TCF dispersion curves for (**a**) SiO_2_/AlN/Diamond multilayer membrane, and (**b**) AlN/Diamond bi-layer membrane with SiO_2_ ribbons for h_SiO2_ = 400 nm and h_Diamond_ = 500 nm.

**Figure 8 sensors-23-06284-f008:**
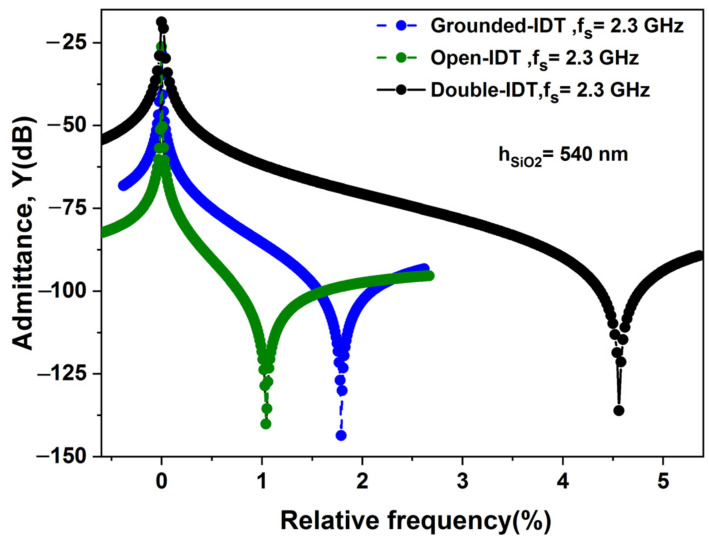
Simulated evolution of the admittance as a function of the relative frequency for three electrode topologies: IDT/AlN/Metal, IDT/AlN and IDT/AlN/IDT for h_SiO2_ = 540 nm and Kh_AlN_ = 2.1.

**Figure 9 sensors-23-06284-f009:**
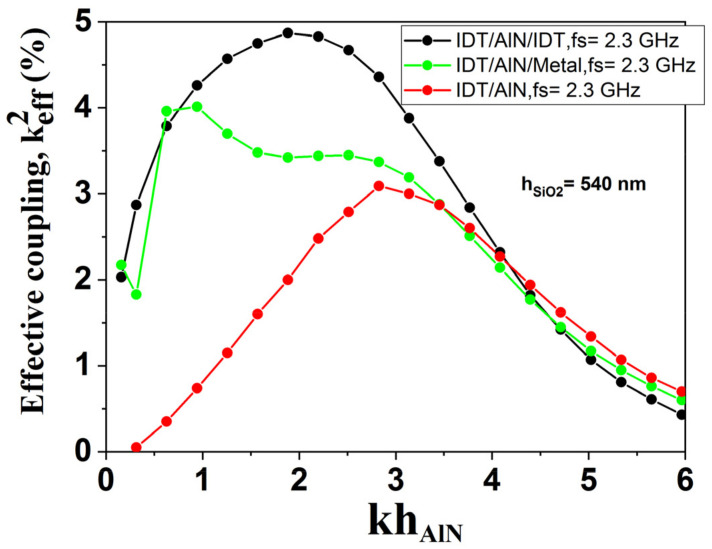
Simulated evolution of $k_{\mathrm{eff}}^{2}$ for three electrode topologies: IDT/AlN/Metal, IDT/AlN and IDT/AlN/IDT using AlN membrane with SiO_2_ ribbons of thickness h_SiO2_ = 540 nm.

**Table 1 sensors-23-06284-t001:** Material parameters of AlN, SiO_2_ and diamond used in the calculations.

Constants	Symbol	AlN [42]	SiO_2_ [42]	Diamond [43]
Stiffness constant (GPa)	C_11_	345	78.5	78.5
	C_12_	125	16.1	16.1
	C_13_	120	16.1	16.1
	C_33_	395	78.5	78.5
	C_44_	118	31.2	31.2
	C_66_	110	31.2	31.2
Temperature coefficient of elastic	TC_11_	−0.37	2.39	−0.14
Constant (10^−4^/°C)	TC_13_	−0.018	5.84	−0.57
	TC_33_	−0.65	2.39	−0.14
	TC_44_	−0.5	1.51	−0.125
Density (Kg/m^3^)	ρ	3260	2210	3510
Piezoelectric constant(C/m^2^)	e_15_	−0.48	-	-
	e_31_	−0.58	-	-
	e_33_	1.55	-	-
Temperature coefficient of mass density	T_ρ_	−14.69	−1.65	−3.6
(10^−6^/°C)				
Dielectric constant (10^−11^F/m)	ε_11_	8	3.32	5.7
	ε_33_	9	3.32	5.7

**Table 2 sensors-23-06284-t002:** Advantages and disadvantages are observed in the range of kh_AlN_ [0.6–2.6] for different device structures.

Device Structure	Transducer Configurations	Advantages	Disadvantages
SiO_2_/AlN bi-layer membrane	electrode topologies (A)	The fabrication process of continuous SiO_2_ film is a little simpler, the fabrication process of a single electrode is a little simpler	low phase velocity, bad temperature stability
	electrode topologies (B)	The fabrication process of continuous SiO_2_ film is a little simpler	low phase velocity, bad temperature stability, the fabrication process of the electrode is less difficult
	electrode topologies (C)	The fabrication process of continuous SiO_2_ film is a little simpler	low phase velocity, bad temperature stability, the fabrication process of an IDT electrode is a little complicated
AlN membrane with SiO_2_ ribbons	electrode topologies (A)	high phase velocity, excellent temperature stability, the fabrication process of a single electrode is a little simpler	―
	electrode topologies (B)	high phase velocity, excellent temperature stability, K^2^ = 3%	The fabrication process of a metal electrode is less difficult
	electrode topologies (C)	high phase velocity, excellent temperature stability, K^2^ = 4%	The fabrication process of IDT electrode is a little complicated
SiO_2_/AlN/Diamond multilayer membrane	electrode topologies (A)	The fabrication process of continuous SiO_2_ film is a little simpler, the fabrication process of a single electrode is a little simpler	low phase velocity, bad temperature stability
	electrode topologies (B)	The fabrication process of continuous SiO_2_ film is a little simpler	low phase velocity, bad temperature stability, the fabrication process of the electrode is less difficult
	electrode topologies (C)	The fabrication process of continuous SiO_2_ film is a little simpler	low phase velocity, bad temperature stability, the fabrication process of an IDT electrode is a little complicated
SiO_2_/AlN/Diamond with SiO_2_ ribbons	electrode topologies (A)	high phase velocity, excellent temperature stability, the fabrication process of a single electrode is a little simpler	―
	electrode topologies (B)	high phase velocity, excellent temperature stability, K^2^ = 3.8%	The fabrication process of a metal electrode is less difficult
	electrode topologies (C)	high phase velocity, excellent temperature stability, K^2^ = 5.2%	The fabrication process of an IDT electrode is a little complicated

## Data Availability

Not applicable.

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
