# Peer review of "Contribution of Ribbon-Structured SiO2 Films to AlN-Based and AlN/Diamond-Based Lamb Wave Resonators"

_sensors, 2023, doi:10.3390/s23146284_

Round 1

Reviewer 1 Report

The experimental study should be added to the manuscript.

No comment.

Reviewer 2 Report

M. Mohammed et al., have studied the characteristics of S0 Lamb wave mode resonators based on AlN membranes combined with SiO2 thin films theoretically. Although there have been a lot of theoretical studies on AlN-based SAW resonating structures, the theme and concept of this work are good. However, the authors need to clarify some issues before it can be accepted in the Sensors journal.

(1)   Please explain the AlN-based piezoelectric materials by comparing their advantages and disadvantages with the other piezoelectric materials. The introduction needs to further enlarges for the better understanding.

(2)   Authors should please give a complete insight for considering the structure A, B, and C, with a detailed and elaborated explanation of differences between the structures such as the merits and demerits table as a supplement, which would be more useful for the readers.

(3)   How about the operating frequency of the as proposed devices? Is it also vary along with K2 and TCF of the resonator structures.

(4)   From reference 36 we didn’t find the information of SiO2 ribbons deposition. Please explain the experimental process. How do the authors estimate the thickness of SiO2 film and SiO2 nanoribbons? In case if the authors have used SiO2 ribbons only for simulation purposes, then how about their physical and optoelectrical properties? What are the length and width of SiO2 ribbons, since their geometrical structure also can induce the device resonating characteristics?

Reviewer 3 Report

the author present a discussion of the device structure proposed by the authors,  I have some suggestions as follows

1, through the author present several structures for the SAW devices, and discussed their K2 and phase velocity, what the physical mechanisin of this ? this should be disscussed;

2, the material parameter used in this work should be presented in the manuscript;

3, how si the Q value and the impedance curve of the devices?

4, How to realize the fabricaiton of such kind of structure should be disscussed. the FEM simulation is such easy, however, the fabrication is so difficult.

the author present a discussion of the device structure proposed by the authors,  I have some suggestions as follows

1, through the author present several structures for the SAW devices, and discussed their K2 and phase velocity, what the physical mechanisin of this ? this should be disscussed;

2, the material parameter used in this work should be presented in the manuscript;

3, how si the Q value and the impedance curve of the devices?

4, How to realize the fabricaiton of such kind of structure should be disscussed. the FEM simulation is such easy, however, the fabrication is so difficult.

Round 2

Reviewer 1 Report

No comments.

Reviewer 3 Report

I have no comment at this stage, the paper can be accepted.